Forecasting the flooding dynamics of flatwoods salamander breeding wetlands under future climate change scenarios

Chandler Houston C. hchandler@oriannesociety.org 1 2
Caruso Nicholas M. 1
McLaughlin Daniel L. 3
Jiao Yan 1
Brooks George C. 1
Haas Carola A. 1
1 Department of Fish and Wildlife Conservation, Virginia Polytechnic Institute and State University (Virginia Tech) , Blacksburg , VA , United States of America
2 The Orianne Society , Tiger , GA , United States of America
3 Department of Forest Resources and Environmental Conservation, Virginia Polytechnic Institute and State University (Virginia Tech) , Blacksburg , VA , United States of America
Brannelly Laura
Electronic publication date: 2023 Sep 19
Publication date: 2023
Volume: 11
Electronic Location ID: e16050
Received 2023 May 22; Accepted 2023 Aug 16
Copyright: ©2023 Chandler et al.
Copyright year: 2023
Copyright holder: Chandler et al.
License: This is an open access article distributed under the terms of the Creative Commons Attribution License, which permits unrestricted use, distribution, reproduction and adaptation in any medium and for any purpose provided that it is properly attributed. For attribution, the original author(s), title, publication source (PeerJ) and either DOI or URL of the article must be cited.
License URL: https://creativecommons.org/licenses/by/4.0/

Keywords: Ambystoma bishopi, Amphibians, Conservation, Ephemeral wetlands, Hydrology, Management, Water level monitoring

Funding: Department of Defense Strategic Environmental Research and Development Program RC-2703 Initial water level monitoring was supported by the Florida Fish and Wildlife Conservation Commission’s Aquatic Habitat Restoration and Enhancement program Intramural research program of the U.S. Department of Agriculture, National Institute of Food and Agriculture, McIntire-Stennis project VA-136640 Funding was provided by the Department of Defense Strategic Environmental Research and Development Program (RC-2703), and initial water level monitoring was supported by the Florida Fish and Wildlife Conservation Commission’s Aquatic Habitat Restoration and Enhancement program. This research was supported by the intramural research program of the U.S. Department of Agriculture, National Institute of Food and Agriculture, McIntire-Stennis project (VA-136640). The findings and conclusions in this publication have not been formally disseminated by the U. S. Department of Agriculture and should not be construed to represent any agency determination or policy. The funders had no role in study design, data collection and analysis, decision to publish, or preparation of the manuscript.

==============================
Ephemeral wetlands are globally important systems that are regulated by regular cycles of wetting and drying, which are primarily controlled by responses to relatively short-term weather events (e.g., precipitation and evapotranspiration). Climate change is predicted to have significant effects on many ephemeral wetland systems and the organisms that depend on them through altered filling or drying dates that impact hydroperiod. To examine the potential effects of climate change on pine flatwoods wetlands in the southeastern United States, we created statistical models describing wetland hydrologic regime using an approximately 8-year history of water level monitoring and a variety of climate data inputs. We then assessed how hydrology may change in the future by projecting models forward (2025–2100) under six future climate scenarios (three climate models each with two emission scenarios). We used the model results to assess future breeding conditions for the imperiled Reticulated Flatwoods Salamander (Ambystoma bishopi), which breeds in many of the study wetlands. We found that models generally fit the data well and had good predictability across both training and testing data. Across all models and climate scenarios, there was substantial variation in the predicted suitability for flatwoods salamander reproduction. However, wetlands with longer hydroperiods tended to have fewer model iterations that predicted at least five consecutive years of reproductive failure (an important metric for population persistence). Understanding potential future risk to flatwoods salamander populations can be used to guide conservation and management actions for this imperiled species.

Introduction

Ephemeral wetlands are globally important ecosystems, contributing to landscape scale hydrological processes and nutrient cycles (McLaughlin, Kaplan & Cohen, 2014; Capps, Berven & Tiegs, 2015; Cohen et al., 2016) and providing critical habitat for diverse and productive communities (Galatowitsch & Van der Valk, 1996; Kirkman et al., 1999; Jenkins, Grissom & Miller, 2001; Gibbons et al., 2006; Hunter & Lechner, 2017). Yet, ephemeral wetlands are imperiled, with an estimated global loss of approximately 50% and accelerated loss in tropical and subtropical areas and in some arid regions (Moser, 1996; OECD, 1996; Davidson, 2014). In the United States, total wetland area has been reduced by approximately 50% since the 1780s, with losses along the coast of the Gulf of Mexico constituting 80% of all wetland losses (Dahl, 1990). While habitat change, especially conversion to agriculture, accounts for the majority of loss across all wetland types (Frayer et al., 1983), climate change represents a future threat to ephemeral wetlands via increases in temperature and changes in precipitation patterns (Dahl, 2011; Zhu et al., 2017). Efforts to protect and restore ephemeral wetlands, therefore, will require an understanding of how climate change may affect wetland hydrologic regimes and associated functions (Erwin, 2009).

In the southeastern United States, climate projections suggest that there will be increased drought severity and frequency (Karl, Melillo & Peterson, 2009), increased rates of evapotranspiration through higher temperatures (Sun et al., 2002; Ingram et al., 2013), and a greater amount of total rainfall that arrives either less predictably or at different times of the year (Mulholland et al., 1997; Karl, Melillo & Peterson, 2009; Ingram et al., 2013). All of these climatic shifts, coupled with features of the local environment and broader landscape (Hayashi & Rosenberry, 2002; Winter & LaBaugh, 2003; Lu et al., 2009; Chandler et al., 2017; Jones et al., 2018), have the potential to alter wetland hydrologic regimes in the region, with cascading effects on water quality, carbon storage, and wildlife populations (Zedler & Kercher, 2005; Davis et al., 2019).

The southeastern United States is a hotspot for amphibian diversity, with many species reliant on ephemeral wetlands to complete their life cycle (Blaustein et al., 2010; Walls et al., 2013; Greenberg et al., 2015). Amphibians often favor ephemeral wetlands for breeding habitat owing to greater reproductive success in the absence of predatory fish populations (Skelly, 1997). However, reproductive failure resulting from wetland drying is common, and altered hydrologic regimes may permanently degrade the conditions necessary to support successful breeding cycles (Brooks, 2009; Yang & Rudolf, 2010; Benard, 2015). As such, amphibians are of particular concern when assessing the vulnerability of different taxa to climate-induced shifts in wetland hydrologic regime (Pounds, 2001; Carey & Alexander, 2003; Blaustein et al., 2010; Walls et al., 2013; Greenberg et al., 2015). In the context of managing imperiled species, failure to consider hydrologic alterations may lead to overly optimistic persistence probabilities derived from population viability analyses or present unexpected challenges to recovery efforts.

Here, we develop a hydrologic model for ephemeral wetlands used as breeding habitat by the federally endangered Reticulated Flatwoods Salamander (Ambystoma bishopi; U.S. Fish and Wildlife Service, 2020). Flatwoods salamanders have declined throughout their range due to widespread loss of their pine flatwoods habitat to urbanization, agriculture, and plantation forestry and the degradation of remaining suitable breeding wetlands, mainly through fire suppression and exclusion (Bishop & Haas, 2005; Palis & Hammerson, 2008; O’Donnell et al., 2017). This degradation of wetland habitat can lead to a variety of conditions that are unsuitable for larval growth and development (e.g., reduced hydroperiod and low dissolved oxygen levels; (De Szalay & Resh, 1997; Skelly, Freidenburg & Kiesecker, 2002; Huxman et al., 2005; Gorman, Bishop & Haas, 2009; Sacerdote & King, 2009; Shulse et al., 2012; Jones et al., 2018). While enhanced vegetation management and prescribed fire application have improved some of these issues, there is substantial concern that climate change will continue to negatively impact flatwoods salamander populations, even in well-managed landscapes (Chandler et al., 2016; U.S. Fish and Wildlife Service, 2020). Using water level data from 35 wetlands within the Gulf Coastal Plain of the southeastern U.S., we sought to predict hydrologic regimes across a series of climate projections to understand future suitability of study wetlands for larval flatwoods salamander survival and metamorphosis. Our research objectives were as follows: (1) model present-day dynamics of ephemeral wetlands, (2) project wetland hydrologic regimes onto future (2025–2100) climate space, and (3) evaluate the impact of future hydrologic regimes on flatwoods salamander breeding potential. Our findings shed light on the key drivers of hydrologic dynamics in the study region and carry implications for flatwoods salamander recovery efforts.

Materials & Methods

Study sites

We conducted our work on Eglin Air Force Base (Eglin, FL, USA) in the Florida panhandle. Access to field sites was approved by the US Fish and Wildlife Service and Jackson Guard (Eglin’s Natural Resources Division; Cooperative Agreement Number F14AC00068). Eglin is a large military installation (>187,000 ha) located in Okaloosa, Walton, and Santa Rosa Counties and consists primarily of actively managed longleaf pine forests. The wetlands studied here have been the subject of ongoing research to understand the variation in hydrologic regime (e.g., Chandler et al., 2016; Chandler et al., 2017) across the landscape with the goal of managing and conserving valuable breeding habitat for flatwoods salamanders (see Chandler et al., 2017 for additional study site details). Extensive habitat management in both uplands and wetlands has made Eglin one of the few remaining strongholds for flatwoods salamanders (Gorman, Haas & Himes, 2013; U.S. Fish and Wildlife Service, 2020).

Water level data

As part of our long-term work on flatwoods salamanders, we previously installed water level monitoring wells at the approximate deepest point in 35 wetland basins on Eglin. All wetlands were located within 20 km of one another, and wetland areas ranged from 0.19–20.92 ha, although all but two wetlands were less than six hectares. Wells were installed from November 2014 through December 2017 such that we had between 1,061 and 2,338 daily measures of water level for each well (see Table S1 for more details). Each well consisted of a 3.8 diameter screened PVC pipe 1 m below ground with a HOBO U20 pressure transducer (Onset Computer Corporation, Bourne, MA) at the bottom of each well that recorded total pressure and temperature at 15-minute intervals. Total pressure data were corrected for barometric pressure variation using pressure sensors installed following methods in McLaughlin & Cohen (2011) at two locations within 9 km of any given wetland (Table S1). For each wetland, we used the resulting 15-min water level data to calculate mean daily water level (mm; positive indicating water levels above ground surface), along with the timing and duration of flooded conditions.

Current climate data

To determine precipitation inputs, we installed four rain gauges (HOBO Data Logging Rain Gauge RG3-M) such that the distance between any of the 35 wells and a rain gauge was no more than 4 km (Table S1). Three of the rain gauges were installed on 20 November 2014 and the fourth was installed on 19 June 2018. We used data from these gauges to calculate daily rainfall (mm), cumulative rainfall over the previous seven days (mm), and the number of days since the last rain event.

We obtained daily minimum and maximum temperature (°C) data from the Florida Automated Weather Network (FAWN) over the course of our study period using the Jay weather station (Station ID 110), which is located 32–58 km from each wetland. We calculated daily potential evapotranspiration (PET, mm) from minimum and maximum temperatures using the Hargreaves-Samani method (Hargreaves & Samani, 1985). Lastly, we calculated the 12-month standardized precipitation-evapotranspiration index (SPEI) for each well from 1 January 1981 to 31 May 2022. The SPEI is useful for assessing drought severity (WMO, 2006) and quantifying and comparing water balances across locations (Stagge et al., 2014). To calculate monthly SPEI, we used daily minimum and maximum temperature data obtained from PRISM Climate group (Oregon State University, http://prism.oregonstate.edu, created 26 Oct 2022), which were necessary because the FAWN data do not have a long enough time series for the SPEI calculation. For the same time period, we used the Hargreaves method (Hargreaves, 1994) to calculate monthly PET, which was used for calculating SPEI.

Hydrologic model

We developed a statistical model to predict daily water levels at each of the 35 wells using a Bayesian first order autoregressive fixed effect model, which was based on visual inspection of partial autocorrelation plots for each well. For each wetland basin, the water level of a given day was modeled as a function of the first order autoregressive term (AR), the amount of rain received during the previous day (Precip), and their interaction (Precip:AR). Models also included linear and quadratic terms for PET, the sum of the rain received in the previous seven days (WeekPrecip), and SPEI. We also included interactions between SPEI and the amount of rain received during the previous day (SPEI:Precip), and the interaction between SPEI, the first order autoregressive term, and the amount of rain received during the previous day (SPEI:AR:Precip). Therefore, for each of the 35 well-specific models, we estimated 11 parameters, including an error term, as shown in the following equation. (1) WaterLevelt=α+βAR+βPrecip+βPET+βPET2+βWeekPrecip+βSPEI+βPrecip:AR+βSPEI:Precip+βSPEI:AR:Precip+ɛ.

We selected these variables to model hydroregimes because of typically strong ephemeral wetland responses to regional climate forcing (Brooks, 2004; Greenberg et al., 2015; Lee et al., 2015). We defined water inputs though direct precipitation effects and parameters that included precipitation within an interaction. The interaction between precipitation and the autoregressive term is useful for defining stage-dependent inputs of rainfall. We included linear and quadratic terms for PET to define short-term (i.e., daily) drawdown of water caused by evapotranspiration because this relationship was nonlinear at low PET values. Lastly, we included SPEI and parameters that include SPEI within an interaction to define wetland response to long-term drought conditions.

For each of the 35 well-specific models, we assigned vague priors following a normal distribution (mean = 0; variance = 100) to all fixed effects, with the error term assumed to follow a normal distribution with mean = 0 and the prior of variance to follow a uniform distribution (min = 0, max = 100). We fit each model using Markov chain Monte Carlo (MCMC) simulations, generating three chains, each with 400,000 total iterations and a thinning rate of 100 (Kéry & Royle, 2016). We used an adaptation phase of 1,000 and discarded 300,000 burn-in iterations, which retained 2,000 iterations for each chain (6,000 total samples) to estimate posterior distributions. We examined traceplots of parameters for adequate mixing among chains and the Gelman–Rubin’s R ˆ statistic to evaluate model convergence (Gelman, 2004). We assessed model predictive ability using a posterior predictive check based on the Bayesian P-value (Kéry, 2010; Link & Barker, 2010). We evaluated parameter significance based on the overlap of 95% highest posterior density with zero.

Model validation

For each well-specific model, we used 75% of the available data to train the model and 25% for testing. Because models were autoregressive, rather than using random data points, the training dataset included consecutive data starting at a random position within the first 25% of the data and consisted of 75% of these data while the testing dataset consisted of the remaining 25% of the data. To determine how well each model predicted the measured water level of our testing and training datasets, we calculated the normalized root mean squared error (NRMSE) for each of the 6,000 iterations, which is the root mean squared error divided by the range of measured water levels within each basin to account for differences in scale among basins. To determine how well each model predicted flooded conditions, we calculated the proportion of days in our testing and training dataset, for each iteration, that were predicted correctly to either have standing water or not.

Predicting hydrologic regime from future climate data

We obtained downscaled climate data for the years 2025–2100 using Localized Constructed Analogs (LOCA; Pierce, Cayan & Thrasher, 2014), which have a 6 × 6 km resolution. For each well, we obtained daily estimates of precipitation (mm), minimum temperature (°C), and maximum temperature (°C) for each of three global circulation models (GCM): Hadley Centre Global Environment Model 2 Earth Systems (HadGEM2-ES), Hadley Centre Global Environment Model 2 Carbon Cycle (HadGEM2-CC), and the Community Climate System model version 4 (CCSM4). For each model, we used two different projections for representative concentration pathways (RCP): 4.5 assumes a peak in carbon emissions in 2040 (Thomson et al., 2011), and 8.5 assumes emissions will continue to increase throughout the 21st century (Riahi et al., 2011). Therefore, for each model, we were able to assess future wetland hydrologic regimes under six different climate scenarios. For each of the six scenarios, we used the daily temperature and precipitation variables to calculate the same climate metrics (i.e., PET, SPEI, sum of rain over the last seven days) as the current data and predicted daily water level for each well using model parameter estimates from their respective posterior distributions.

To predict future hydrologic regimes and associated habitat suitability for flatwoods salamanders, we combined the LOCA climate data and the parameter posterior distributions from our statistical models. For each breeding season from 2025–2100, we determined the length (number of days) of the maximum hydroperiod within a given season, as well as the day in which each wetland became flooded. We defined the flatwoods salamander breeding season as occurring between 1 November and 31 May of the following year (Palis, 1996). Additionally, within a given posterior draw, we determined the number of consecutive breeding seasons that did not contain a hydroperiod of at least 15 weeks (105 days). While metamorphosis in flatwoods salamanders has occurred in as little as 11 weeks (77 days) and may take as long as 18 weeks (126 days; Palis, 1995), 15 weeks is a conservative estimate for the typical length of wetland filling required to allow larval development through metamorphosis (Brooks et al., 2020; Haas, 2010–2020, unpublished data) . Lastly, for each of the maximum hydroperiod estimates, we determined the date the wetland filled as the number of days since 1 November for the respective breeding season.

All analyses were performed in program R (version 4.1.1; R Core Team, 2021). We used the SPEI (Baguería & Vicente-Serrano, 2017) and Evapotranspiration (Guo, Westra & Peterson, 2020) packages for calculating SPEI and PET respectively, and the jagsUI package (Kellner, 2021) to call JAGS (Plummer, 2003), from Program R for MCMC analyses.

Results

Model fitting

Our dataset consisted of 72,266 daily observations of wetland water level across 35 monitoring wells (range: 795–1,753 training observations per well; Table S1). These data covered a range of hydrologic conditions, both among wetlands (e.g., mean across wetland hydroperiod during the flatwoods salamander breeding season ranged from 37–142 days) and across years (e.g., mean annual breeding season hydroperiod across all wetlands ranged from 31–168 days). All MCMC chains showed good mixing, and R ˆ values were between 1.000 and 1.003, indicating model convergence. The posterior predictive check indicated that each model fit the data well (Bayesian P-value range: 0.49–0.51).

Our models generally showed good predictability of water level data, with the median training NRMSE ranging from 0.01 to 0.04 and the testing NRMSE ranging from 0.13 to 0.35 (Figs. S1A and S1B). Variance in NRMSE was always <0.01 for training and testing datasets and generally showed a decreasing trend with the number of training or testing datapoints. The median proportion of days correctly predicted to either have or not have surface water ranged from 0.94–0.99 for training datasets and from 0.56–0.93 for testing datasets (Figs. S1C and S1E). Averaged across all model iterations, the percentage of testing days predicted to be dry when the wetland was flooded ranged from 1.2–39% per wetland, while the percentage of testing days predicted to be flooded when the wetland was dry ranged from 0.3–29.4% per wetland. Some wetlands were more likely to misclassify wet or dry days, but others were equally likely to misclassify either condition. The variance in proportion of days correctly predicted to have surface water present ranged from <0.001–0.01 and from 0.12–2.40 for training and testing datasets, respectively (Fig. S1D and S1F).

Model parameters

The mean value for the first order autoregressive coefficient was positive and slightly less than one for all wetland basins. We found that daily precipitation had the largest magnitude effect, when compared to all other parameters, on wetland water level (Fig. 1; Table S2). The amount of precipitation over the previous seven days had a smaller positive effect than daily precipitation on water level. The model intercept (α), the SPEI, the quadratic term for PET, and the interactions between SPEI and daily rainfall and daily rainfall, autoregressive term, and SPEI had both negative and positive effects depending on the model. Finally, the effects of PET and the interaction between precipitation and the autoregressive term tended to have negative effects on daily water level (Fig. 1; Table S2).

Figure 1 Median parameter estimates for Bayesian first order autoregressive fixed effect models that predicted daily water levels for wetlands on Eglin Air Force Base, Florida.

Model parameters included an intercept (α), an autoregressive term (βAR), precipitation (βPrecip), potential evapotranspiration (βPET), total precipitation over the previous seven days (βWeekPrecip), the 12-month standardized precipitation-evapotranspiration index (βSPEI), the associated interactions and quadratic effects, and an error term (ɛ). Points represent the median value for each of the 35 monitoring wells included in this study, and the vertical line indicates whether effects were positive or negative. For example, precipitation, weekly precipitation, and the autoregressive term had positive effects across all models, while PET tended to have a negative effect on daily water level. Most other terms had both positive and negative effects, depending on the specific wetland.

Future predictions

Future projections revealed suitable hydroperiods for flatwoods salamander larval development (i.e., at least 15 consecutive weeks or 105 consecutive days of surface water during the breeding season) in at least some years across all climate scenarios (Figs. 2 and 3). When comparing the various climate scenarios, the CCSM4 model predictions typically had fewer years with suitable hydroperiods compared to the HadGEM2-CC or HadGEM2-ES model (Figs. 2 and 3). For date of wetland filling, wetland basins generally showed a tendency towards earlier fill dates (Fig. S2). For both hydroperiod and date of wetland filling, there was considerable variation among years and iterations for all wetland basins (Figures 2–3 and Fig. S2). Lastly, across all wetlands and climate scenarios examined in this study, we found substantial variability in their general suitability for flatwoods salamander reproduction (Fig. 4). We found that wetlands experiencing longer hydroperiods, on average, were less likely to have model iterations of at least five consecutive years of reproductive failure (i.e., five consecutive years with a hydroperiod less than 105 days). This metric is thought to be important for flatwoods salamander population persistence (Palis, Aresco & Kilpatrick, 2006). Some wetlands did have a high probability (>75%) of extended recruitment failures or had a low number of years with a suitable hydroperiod, indicating they are likely unsuitable for flatwoods salamanders. Overall, most wetland basins had suitable hydroperiods for flatwoods salamanders in at least 50% of future years (Figs. 2 and 4).

Figure 2 Predicted hydroperiods for a single ephemeral wetland on Eglin Air Force Base, Florida.

Probability of a 15-week (105-day) hydroperiod in a single Reticulated Flatwoods Salamander (Ambystoma bishopi) breeding wetland. There have been reports of successful metamorphosis of flatwoods salamanders after an 11-week (77-day) hydroperiod, but 15 weeks is a more conservative estimate of suitable hydroperiod. Predictions were made across three climate models (Hadley Centre Global Environment Model 2 Earth Systems (HadGEM2-ES), Hadley Centre Global Environment Model 2 Carbon Cycle (HadGEM2-CC), and the Community Climate System model version 4 (CCSM4)) and two emission scenarios (RCP4.5: assumes a peak in carbon emissions in 2040; RCP8.5: assumes emissions will continue to increase throughout the 21st century). Across all models and through time, many years appear to likely have suitable conditions for flatwoods salamander reproduction, and there were no strong temporal trends in hydroperiod predictions across models. Hydroperiods were calculated from 1 November to 31 May.

Figure 3 Median predicted hydroperiod from 2025 to 2100 for 35 wetlands on Eglin Air Force Base, Florida.

Lines represent smoothed (using the loess function in R) predictions based on 6,000 iterations of Bayesian first order autoregressive models that each simulated conditions under three global climate model (Hadley Centre Global Environment Model 2 Earth Systems (HadGEM2-ES), Hadley Centre Global Environment Model 2 Carbon Cycle (HadGEM2-CC), and the Community Climate System model version 4 (CCSM4)) and two emission scenario (RCP4.5: assumes a peak in carbon emissions in 2040; RCP8.5: assumes emissions will continue to increase throughout the 21st century) combinations (six total scenarios). Plots show overall variability in model predictions across all study wetlands. Hydroperiods were calculated across the Reticulated Flatwoods Salamander (Ambystoma bishopi) breeding season (1 November to 31 May), and 105 days represents a conservative estimate for the time needed for successful metamorphosis. Wetlands are grouped based on general hydroperiod conditions: long (A), medium (B), or short (C), and asterisks at the end of the wetland name indicate wetlands that have been confirmed occupied by flatwoods salamanders during the previous 10 years.

Figure 4 Relationship between wetland hydroperiod and probability of consecutive short-hydroperiod years for wetlands on Eglin Air Force Base, Florida.

Relationship between the median number of Reticulated Flatwoods Salamander (Ambystoma bishopi) breeding seasons (1 November to 31 May) with an at least 15-week hydroperiod from 2025–2100 versus the proportion of iterations for which a given wetland had at least five consecutive years without a hydroperiod of 15 weeks. A 15-week hydroperiod is a conservative estimate for the time needed for successful reproduction by flatwoods salamanders, and five consecutive years without reproduction (a hydroperiod shorter than 15 weeks) is a metric thought to be important for population persistence. The vertical (0.75) and horizontal (15) dotted lines represent cutoffs indicating a high probability (i.e., above 75% of proportions or less than 20% of breeding seasons) of extirpation of the salamander population. Predictions were made across three global climate models (Hadley Centre Global Environment Model 2 Earth Systems (HadGEM2-ES), Hadley Centre Global Environment Model 2 Carbon Cycle (HadGEM2-CC), and the Community Climate System model version 4 (CCSM4)) and two emission scenarios (RCP4.5: assumes a peak in carbon emissions in 2040; RCP8.5: assumes emissions will continue to increase throughout the 21st century).

Discussion

Here, we present wetland-specific models describing hydrologic regime in ephemeral wetlands embedded within pine flatwoods of the southeastern United States. We show that, by harnessing information contained within multiple years (2014–2022) of daily water level data, models accurately simulated the dynamics of 35 flatwoods salamander breeding wetlands. Projecting across six climate change scenarios, we show that wetlands vary in their response to simulated climate scenarios, with many wetlands exhibiting earlier fill dates in future years. Despite these projected alterations in hydrologic regime, most sites appear to remain hydrologically suitable for flatwoods salamanders. For the handful of wetlands that may become unsuitable, our results can be used to direct appropriate management practices to mitigate climate-induced changes.

Hydrologic regimes were most strongly determined by precipitation patterns and PET rates. Specifically, daily precipitation had the largest positive influence on wetland water level, and linear relationships with PET had the largest negative effect on wetland water level. It is unsurprising that daily precipitation had a large positive effect on water levels as these wetlands are largely supported by precipitation and shallow groundwater dynamics, with minimal surface water connections to other water sources (i.e., often referred to as geographically isolated wetlands; Chandler et al., 2017; Tiner, 2003). Similarly, the effects of vegetation (and thus ET rates), both within and surrounding wetlands, are known to influence groundwater inputs and resulting water levels (Jones et al., 2018). A logical conservation application, therefore, is to devise management strategies designed to artificially manipulate these important determinants of hydrologic regime (e.g., Golladay et al., 2021). Future research should seek to test the effectiveness of shallow groundwater wells or different vegetation management practices (including the restoration of natural fire regimes) in extending wetland hydroperiods (Seigel, Dinsmore & Richter, 2006; Jones et al., 2018).

Our long-term water level data allowed us to accurately model wetland hydrologic regime under a range of both current and future climate conditions. However, accurate predictions were contingent on having multiple years of data, and our dataset included both wet and dry years across all wetlands. The amount of data necessary to construct robust models likely represents a need to capture different water levels under a wide range of climate conditions. Unsurprisingly, annual climate variation is reflected in hydroperiod as these ephemeral wetlands display markedly different hydroperiods from one year to the next (Chandler et al., 2016). Therefore, a single breeding season or year will likely not capture this variation, especially as precipitation events vary in number and magnitude. Additionally, the magnitude of increases in wetland water level to a given amount of precipitation will change under long-term drought (e.g., low values of SPEI) or wet conditions (high values of SPEI), highlighting the need for extended time series to quantify long-term trends.

Although predictive accuracy varied considerably across the 35 instrumented wetlands, this variability was unrelated to the number of training data points used in model fitting. This suggests two things, (1) even the most recently instrumented sites yielded enough data points to accurately predict their dynamics (>1,000 total data points), and (2) residual error in model predictions was due to unmeasured variables, as opposed to insufficient data. Characteristics such as area, shape, hydraulic conductivity, and vegetation structure are all expected to affect wetland hydrologic regime and its response to climate forcing (Brooks, 2005; Jones et al., 2018; Cianciolo et al., 2021). For example, the large variation in the degree to which daily precipitation increases water levels at our study wetlands likely results from differences in canopy interception, wetland bathymetry, and storm event surface runoff (Brooks, 2005). Further, larger wetlands have been shown to exhibit lower recession rates and longer hydroperiods compared to smaller wetlands (e.g., Vanschoenwinkel et al., 2009; Chandler et al., 2016; Chandler et al., 2017). It is likely that other factors, such as wetland bathymetry (Haag, Lee & Herndon, 2005) and landscape position, may also affect wetland hydrologic response to climate variables, and as such, incorporating other wetland characteristics could improve the predictive accuracy of these models and provide additional insights into an individual wetland’s sensitivity to climate change.

Changes in the timing or duration of wetland filling, like those our models project, can negatively impact several aspects of amphibian reproduction (Parmesan, 2006; Li, Cohen & Rohr, 2013). During the fall and early winter, adult flatwoods salamanders migrate to wetlands to lay their eggs in dry wetland basins (Anderson & Williamson, 1976; Palis, 1996). Embryos begin to develop terrestrially but do not hatch until inundated by rising water levels. Although eggs can persist in dry basins for up to two months, they risk mortality from desiccation or freezing if exposed for too long (Anderson & Williamson, 1976). After inundation, larvae can take between 11 to 18 weeks to metamorphose into terrestrial adults (Palis, 1995). Therefore, reproductive success is dependent upon both the timing of when a wetland fills at the beginning of the breeding season as well as the length of time a wetland is flooded. Reassuringly, our future projections of hydrological suitability suggest that most of the currently monitored wetlands will remain suitable for flatwoods salamander larval development and metamorphosis. Additionally, the strong influence of PET on water level suggests a potential management strategy involving the intensive removal of woody vegetation (e.g., Liu et al., 2020). Indeed, many of these wetlands, as they are either currently occupied or potentially suitable for flatwoods salamanders, have been the focus of habitat management in the form of physical removal and chemical treatment (i.e., herbicide and fire) of wetland shrubs (Gorman, Haas & Himes, 2013), presenting an opportunity to quantify the additional benefits of vegetation removal on habitat quality.

Conclusions

There is a growing body of literature highlighting observed and potential impacts of climate change on amphibian species in the southeastern United States (e.g., Todd et al., 2011; Greenberg et al., 2015; Walls et al., 2019). Our model results do not suggest an immediately worrisome scenario for flatwoods salamanders when considering breeding wetland hydrology in the coming decades. However, it is important to acknowledge that population viability is contingent on the timing of breeding migrations in relation to environmental conditions and the survival of adults in upland habitats surrounding breeding wetlands (Brooks, 2020). Only by integrating the wetland models presented here with additional phenological and demographic information can we explicitly model flatwoods salamander persistence under future climate change and guide ongoing recovery efforts. More broadly, our approach can be used to discern the relative vulnerability of ephemeral wetlands in the southeastern United States to climate change and devise strategies to safeguard the species that rely on them.

Supplemental Information

Table S1 Additional details about water level monitoring in pine flatwoods wetlands on Eglin Air Force Base, Florida

Click here for additional data file.

Table S2 Additional details of model results for models describing hydrologic regime in pine flatwoods wetlands on Eglin Air Force Base, Florida

Median, lower, and upper 95% highest density posterior distributions for model parameters and wetland basins. Model parameters included an intercept (α), an error term (σ), precipitation (βRAIN), an autoregression term (βAR1), total precipitation over the previous seven days (βWEEKRAIN), the 12-month standardized precipitation-evapotranspiration index (βSPEI), potential evapotranspiration (βPET), and associated interactions and quadratic effects.

Click here for additional data file.

Figure S1 Plots describing performance of models predicting water levels in ephemeral wetlands on Eglin Air Force Base, Florida

(A, B) Model accuracy for predicted water levels was estimated by normalized root mean squared error (NRMSE) and (C, D, E, F) for the prediction of length of presence of surface water by the proportion of days that correctly predicted to either have or not have water. Results are divided by the number of training (A, C, D) or testing (B, E, F) data points (daily observations) used in the model. Error bars in A, B, C, and E represent the upper and lower 95% highest density posterior and points represent median values, while points in D and F are variance estimates. Note that the y-axis scale is different for all six panels.

Click here for additional data file.

Figure S2 Median predicted date of basin filling from 2025 to 2100 for 35 wetlands on Eglin Air Force Base, Florida

Lines represent smoothed (using the loess function in R) predictions based on 6,000 iterations of Bayesian first order autoregressive models that each simulated conditions under three global climate model (Hadley Centre Global Environment Model 2 Earth Systems [HadGEM2-ES], Hadley Centre Global Environment Model 2 Carbon Cycle [HadGEM2-CC], and the Community Climate System model version 4 [CCSM4]) and two emission scenario (RCP4.5: assumes a peak in carbon emissions in 2040; RCP8.5: assumes emissions will continue to increase throughout the 21st century) combinations (six total scenarios). Plots show overall variability in model predictions across all study wetlands. Fill dates were calculated relative to the Reticulated Flatwoods Salamander (Ambystoma bishopi) breeding season (1 November to 31 May) as days after November 1. Wetlands are grouped based on general fill date conditions: variable across approximately three months (A), two months (B), or one month (C) after November 1, and asterisks at the end of the wetland name indicate wetlands that have been confirmed occupied by flatwoods salamanders during the previous 10 years.

Click here for additional data file.

File S1 Water level monitoring data from 35 pine flatwoods wetlands on Eglin Air Force Base, Florida

Click here for additional data file.

We thank the following individuals for their assistance with this project: T. Anderson, J. Barichivich, J. Davenport, C. Ewers, R. Felix, S. Goodman, T. Gorman, B. Hagedorn, M. Holden, J. Johnson, K. Jones, K. O’Donnell, V. Porter, J. Preston, B. Rincon, and S. Walls. A myriad of seasonal technicians assisted with fieldwork over the course of this project. The manuscript was improved by comments from L. Brannelly, T. Hossie, and two anonymous reviewers. Logistical assistance was provided by the Department of Fish and Wildlife Conservation at Virginia Tech, Jackson Guard (Eglin Air Force Base’s natural resources division), and the U.S. Fish and Wildlife Service Panama City Field Office.

Additional Information and Declarations

Competing Interests

Author Contributions

Field Study Permissions

Data Availability

The authors declare there are no competing interests.

Houston C. Chandler conceived and designed the experiments, prepared figures and/or tables, authored or reviewed drafts of the article, and approved the final draft.

Nicholas M. Caruso conceived and designed the experiments, performed the experiments, analyzed the data, prepared figures and/or tables, authored or reviewed drafts of the article, and approved the final draft.

Daniel L. McLaughlin conceived and designed the experiments, authored or reviewed drafts of the article, and approved the final draft.

Yan Jiao performed the experiments, analyzed the data, authored or reviewed drafts of the article, and approved the final draft.

George C. Brooks conceived and designed the experiments, authored or reviewed drafts of the article, and approved the final draft.

Carola A. Haas conceived and designed the experiments, authored or reviewed drafts of the article, and approved the final draft.

The following information was supplied relating to field study approvals (i.e., approving body and any reference numbers):

Fieldwork and access to field sites were approved by the U.S. Fish and Wildlife

Service and Jackson Guard (Eglin Air Force Bases Natural Resources Division).

The following information was supplied regarding data availability:

The raw data are available in the Supplemental Files.

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
