# Peer review of "Forecasting the flooding dynamics of flatwoods salamander breeding wetlands under future climate change scenarios"

_PeerJ, doi:10.7717/peerj.16050_

## Round 0.1 · original submission · Minor Revisions

Thank you for submitting to PeerJ. Your manuscript has been reviewed by 3 experts in the field, and all 3 are supportive of this manuscript being published with some minor revisions. The work was clearly laid out, and is important conservation research. Please revise your manuscript and address the specific and thoughtful reviewer feedback. I look forward to reading your revised manuscript

Reviewer 1 ·

Basic reporting

The manuscript is well constructed and conveys important information clearly in a logical order. Citations are relevant. Conclusions follow Results. Results follow Methods. Methods follow hypotheses and objectives.

Experimental design

The authors selected an appropriate and modern statistical modeling approach for the research questions and hypotheses. However, the model is not described in sufficient detail in the Methods. Not all readers will be able to visualize the model structure based on the text in lines 154-163. I strongly recommend either including a Table, like the Tables in the Supplementary materials or a better option would be to present the model formula in the Methods.

Validity of the findings

The authors did a great job of describing the model testing and validation processing. The authors could consider additionally reporting the proportions under and over predicting. This would help to understanding if the incorrect predictions were systematic or stochastic.

Additional comments

In the Discussion, the authors suggest using shallow groundwater wells (Line 303) but previously indicate that these wetlands are predominantly supplied by rainwater (Lines 2096-299 – needs a citation to support this – one of the ones from the Study Site section would be fine). These are not mutually exclusive ideas but the authors need to better explain why shallow groundwater, which is not an important contributor, would be available during periods of drying. I understand why, because I work in both groundwater supplied and precipitation supplied systems, however, when I was new to wetland science, this would have been confusing.

Please see comments for each Figure.
Figure 1. Multi-panel figures are efficient for conveying information, however, the number of panels compresses the information too much. Panels B and D may be eliminated.
Figure 2 needs to be organized by model structure (i.e., from intercept through main effects and culminating in the interactions). The estimates of the interactions depend on the main effects and should be presented that way.
Figure 3 should be oriented to increase size. This is a very useful figure but hard to read.
Figure 4’s panels and Figure 5’s panels are too small as to be unusable.

Raw data can be opened. I recommend labeling Precip as Daily Precip to be clear and consistent. I recommend correcting the spelling of Dailey (sp) PET.

Language is appropriate and clear.

·

Basic reporting

This manuscript meets these criteria.

The paper is very clear and well written. Appropriate literature is cited, and sufficient background context is provided.

Figures will need to be updated with higher-resolution versions at the time of publication. Text size for the axes titles and legend text on Figures 4 and 5 should be increased.

Experimental design

Paper fits the aims and scope of the journal.

The research questions were clearly defined and meaningful. The manuscript clearly addresses the research gap that the authors identified in their introduction.

The experimental design is appropriate, and the effort that went into collecting this data is commendable. The data wrangling and extensive effort to robustly analyze this data is also commendable. There is a very high technical standard met here for the data analysis.

Methods are described clearly. Researchers in this field would be able to replicate what was done here based on the level of detail provided.

Validity of the findings

The results presented in the manuscript are valid, and interpreted robustly. The underlying data appear to be robust, and the analyses appear statistically sound. The authors have controlled for all of the variables they had data for, and have clearly acknowledged the kinds of variables which they did not have which could influence their results and model fit (e.g., likes 319-322). Main conclusions are clearly stated and well-grounded in the well-supported results that the authors presented.

Additional comments

Overall, the authors have presented a very useful analysis that provides important insight into the long-term viability of the salamanders in question. This approach is general enough that it could be translated to evaluate risk in other wetland systems, which makes the paper even more valuable. The paper is clear, well written, and well justified. I have some additional comments that are meant to help make this paper stronger and more accessible to a broad readership.

1) Some of the results presented in the Results section are somewhat qualitative (e.g. "We found that wetlands experiencing longer hydroperiods, on average, were less likely to have model iterations of at least five consecutive years of reproductive failure..." lines 276-274). These could be more quantitatively supported by statistical tests (or even just descriptive statistics).

2) It would be helpful to know to what extent are the wetlands that fall outside of the suitable range always the same wetlands irrespective of the RCP or climate model? I was left wondering whether it was some wetlands in particular that were just consistently poor across all scenarios, or if how many moved from suitable to unsuitable depending on the RCP/Model combination. This could be addressed in the Results section.

3) Most readers are not likely to get much value out of Figure 1. I would suggest that it could be moved into the supplement.

4) Many demographic researchers suggest limiting forecasts to 50 years because predictions that extend well into the future become increasingly unreliable. The authors might consider limiting their predictions and inferences to 2075, opposed to extending it 75 years into the future.

5) Figures 4 and 5 are somewhat overwhelming, especially considering that we have no context on their size, shape or location. Consider how these figures might be simplified. The y-axes are one several different scales, which makes direct comparison among wetlands difficult. Consider standardizing them to a common y-axis range & interval. The authors may also wish to add a line indicting the minimum required hydroperiod for successful breeding (~105 days), or at least state this in the figure caption. Similarly they could indicate a key date cutoff on Figure 5 or in the caption.

6) May readers might find it hard to interpret Figure 6. The authors might consider adding text in the caption, or revising axes titles to help non-specialist readers understand the key take-away message from this figure.

7) If not precluded from doing so due to data sensitivity, a map which depicts the location of the wetlands and includes the locations of the monitoring stations would be helpful.

Reviewer 3 ·

Basic reporting

1) Basic reporting - This manuscript uses hydrologic modeling to look at the suitability of Flatwoods Salamander habitat under a range of climate change scenarios.
a. The manuscript is written clearly and follows the conventions of US science writing.
b. The introduction provides a clear rationale for the study and appropriate citations are provided.
c. The structure conforms to PeerJ standards.
d. Figures are generally high quality. I noted that a couple are very hard to read because of font size and number of panels per page. This is noted in the text. I will let the editors and authors resolve the issue of figure size.
e. An excel file containing raw data was available for review.

Experimental design

2) Experimental design
a. An evaluation of breeding habitat suitability for listed amphibian species using downscaled GCMs falls within the scope of PeerJ.
b. The subject of the research is relevant to long term conservation of the flatwood’s salamander. The research is also relevant to understanding how climate change will affect geographically isolated wetlands.
c. The field portion of the investigation was well conducted. I am not a modeler so I will leave that aspect of the review to others.
d. The methods were very well written. It is rare that a non-modeler can follow the details that go into calibrating and using a hydrologic model. The authors are commended for their efforts at communicating. I believe there is sufficient detail to allow replication.

Validity of the findings

3) Validity of findings
a. This is a thoughtful examination of how climate change might affect geographically isolated wetland hydroperiods and the implications for a listed species. The relevance to conservation of the species is clear.
b. Data are provided and could be reanalyzed if that was someone’s desire. I did not perform any statistical checks on them.
c. The conclusions are well-stated and flow logically from the project rationale and results.

Additional comments

4) Custom checks
a. I have no way of verifying the authors’ field study permits. However, given that they are civilians working on a major military facility with very stringent security, I feel confident that the paperwork was in place.

Annotated reviews are not available for download in order to protect the identity of reviewers who chose to remain anonymous.

---

## Round 0.2 · Minor Revisions

Thank you for resubmitting your manuscript to PeerJ. As stated before, this research is important and well-reported, and deserves to be published.

However, I have to agree with the reviewers in that all 3 said that figures 4 and 5 are overwhelming for the reader, and it seems like you did not adjust these figures for the manuscript. If you are set on showing these exact figures, then they are better suited as a supplement. And in the main text, you could provide a summary figure. It's unclear what the reader is supposed to take away from understanding the patterns in each individual basin, but perhaps a summary of all the basins with similar geology/ecology would be more useful.

Furthermore, the reviewers pointed out that the scales for each basin panel are different, which does make it difficult to understand and the patterns difficult to pick out. While I know that formatting figures in R can be a challenge, you could try reordering the panels so that the basins with the same scale are next to each other. The basin number is arbitrary anyway (as I understand it - they identify a specific basin rather than a continuous variable about the basin) therefore the order does not matter and does cloud the interpretability of the figure. Please readjust these figures, and place them in a supplement.

I also find Figure 3 difficult to grasp. The bars are small, the grey and black are very similar in tone, and the panel labels take up a lot of space but are not particularly useful upon first glance in terms of their meaning. Your figures should be stand-alone and not rely on the reader's deep understanding of the text for them to understand the meaning of the figures. - are the black and the grey bars additive? Because if they are not, then the patterns are relatively clear for the grey bars, but difficult to understand for the black ones - side-by-side might be easier, or even a density curve could be easier to understand. I'm unsure what the key takeaway from this figure is supposed to be, it looks like there are no clear patterns in the probability of a hydroperiod and regardless of the model, there are sometimes high and low probability of a hydroperiod with no patterns across years. If the figure is important to include in the main text, then it should tell a story. Otherwise, if you simply want to show the results of your models, then they should be placed in supplemental.

I agree with the reviewers in what they said about Figure 1 not being useful to the readers in the main text of the paper. However, I do think that it is a perfect example of a supplemental figure.

Figure 2 is difficult to understand without the main text, and as said before, figures should be stand-alone. What is the vertical dotted grey line? I know it is at 0 but what does it broadly indicate in this context? The language used here is specific to the type of model you ran but is going to be difficult to understand for your average PeerJ reader. I would highly encourage you to broaden the language used to describe this figure. You've done a great job with this in the manuscript: eg lines 262-265. Please use the same broadly understood language in the figures.

I agree that you have done an excellent job editing the main text of the manuscript, but I do think some work is needed with the presentation of the figures. They are too complex and detailed for the average reader to broadly understand, and because clear patterns are not presented, it will negatively affect the article's readership. Please consider creating summary figures showing broad patterns across water basins rather than specific predictions for each basin.

---

## Round 0.3 · accepted · Accept

Thank you for making the appropriate changes and/or explaining why you made some of the choices you made in your rebuttal letter. I believe the manuscript is ready for publication. Congratulations!